# New Approaches to the Synthesis and Stabilization of Polymer Microspheres with a Narrow Size Distribution

**DOI:** 10.3390/polym15112464

**Published:** 2023-05-26

**Authors:** Inessa A. Gritskova, Nikolay I. Prokopov, Anna A. Ezhova, Anatoly E. Chalykh, Sergey A. Gusev, Sergey M. Levachev, Vitaly P. Zubov, Vitaly I. Gomzyak, Ivan V. Skopintsev, Alexander N. Stuzhuk, Ivan D. Kovtun, Anton M. Shulgin, Dmitry S. Ivashkevich, German A. Romanenko, Valentin G. Lakhtin, Sergei N. Chvalun

**Affiliations:** 1Department of Chemistry and Technology of Macromolecular Compounds, MIREA—Russian Technological University (RTU MIREA), 119454 Moscow, Russia; 2Frumkin Institute of Physical Chemistry and Electrochemistry, Russian Academy of Sciences (IPCE RAS), Leninskiy Prospekt 31, 119071 Moscow, Russia; 3Federal Research and Clinical Center of Physical-Chemical Medicine of Federal Medical Biological Agency, Malaya Pirogovskaya 1a, 119435 Moscow, Russia; 4Federal State Autonomous Educational Institution of Higher Education “Pirogov Russian National Research Medical University”, Ministry of Health of the Russian Federation, Ostrovitjanova St. 1, 117997 Moscow, Russia; 5Faculty of Chemistry, Lomonosov Moscow State University, GSP-1, 1-3 Leninskiye Gory, 119991 Moscow, Russia; 6National Research Centre “Kurchatov Institute”, Akademika Kurchatova pl. 1, 123182 Moscow, Russia; 7Institute of Petrochemical Synthesis, Russian Academy of Sciences, 29 Leninsky Prospect, 119991 Moscow, Russia

**Keywords:** surfactant, polymeric microspheres, stabilization

## Abstract

This article presents the results of investigations on heterophase polymerization of vinyl monomers in the presence of organosilicon compounds of different structures. On the basis of the detailed study of the kinetic and topochemical regularities of the heterophase polymerization of vinyl monomers, the conditions for the synthesis of polymer suspensions with a narrow particle-size distribution using a one-step method have been determined.

## 1. Introduction

Polymer dispersions with a narrow particle-size distribution are widely used in medicine and biology [1,2,3,4,5,6,7]. They are now firmly established in diagnostics and are used instead of red blood cells as carriers of bioligands to create diagnostic test systems based on antigen–antibody reactions [1]. The surface of polymeric microspheres should contain functional groups (amino, carboxyl, aldehyde, etc.) capable of providing an opportunity for covalent immobilization of bioligands while maintaining natural conformation of the bioligand. Another relevant and important use of polymer microspheres is based on their use as a kind of sorbent for neutrophil extracellular traps, network-like structures ejected into the extracellular space by neutrophil leukocytes upon activation. In some diseases (diabetes mellitus, rheumatoid arthritis, cancer), a large number of neutrophil extracellular traps form in the blood stream, leading to the development of severe complications. In this regard, the problem of determining their number and a method for their removal from the blood is extremely relevant. This problem can be solved with the use of polymer microspheres.

The traditional and most common methods of polymer suspension synthesis are emulsion and suspension polymerizations. The basic data on the mechanism of suspension particle formation and the ways to regulate their size and diameter distribution are described in detail in the literature [1,2,3,4,5,6,7,8,9,10,11,12,13,14,15,16,17,18,19,20,21,22,23,24,25,26,27,28,29,30,31,32].

According to Harkins, Yurzhenko, Smith, and Ewart [3,4,5,6,7,8,9], the particles are formed from the surfactant micelle containing the monomer, after a radical enters it, while the droplets are considered as reservoirs of the monomer. The monomers entered into the particles by diffusion through the aqueous phase, replenishing its flow rate. The width of particle-size distribution was related to the duration of the initial polymerization stage, that is, the formation stage of polymer–monomer particles (PMPs) with diameters of not more than 0.3 µm and with wide particle-size distribution.

## 2. Role of the Microemulsification Process in Particle Formation in Emulsion Polymerization

Despite the Smith–Ewart theory of emulsion polymerization described in the literature [3,4,5,6,7,8,9], it was shown [20,21,22,23] that PMPs could be formed not only from the surfactant micelles but also from the monomer microdroplets formed as a result of quasi-spontaneous microemulsification due to surfactant transfer across the interface according to its solubility in the monomer and in water. The most effective monomer microemulsification is observed when the emulsifier is synthesized directly at the interface as a result of a chemical reaction between a long-chain carboxylic acid dissolved in the monomer and an alkali dissolved in the aqueous phase, or when the emulsifier is introduced into the phase of the emulsion in which it is less soluble [23,31,32]. In these cases, a large number of microemulsion droplets is formed and the initiation of polymerization sets the stage for the formation of PMPs, predominantly from monomer microdroplets. When polymerization is complete, the polymer suspension is characterized by a narrow particle-size distribution.

A comprehensive theoretical analysis of the microemulsification process with an estimate of the size of the formed monomer microdroplets is still lacking. Possible mechanisms of microemulsification are considered [23] to be the short-term creation of a high surfactant concentration near the interfacial boundary and the formation of a microemulsion in accordance with local equilibrium conditions. The resulting monomer microdroplets are thermodynamically unstable, but their lifetime is sufficient for them to participate in the formation of PMPs, so this mechanism of microemulsion formation has been conventionally termed as quasi-equilibrium. Creating the conditions for the formation of PMPs from monomer microdroplets opens up the possibility of regulating the particle diameter and particle-size distribution [23].

It should be noted that the size of monomer microdroplets (~0.2 µm) exceeds the typical size of emulsifier micelles, but it is significantly lower than the size of the droplets that can be obtained by mechanical dispersion of the monomer at a corresponding value of interfacial tension. The mechanism of polymerization in the PMPs formed from microdroplets of this diameter is similar to an emulsion polymerization of the monomer, forming a high-molecular-weight polymer. However, manufacturing of larger particles using this approach is problematic.

## 3. Formation of Polymeric Microspheres with Larger Diameters above 0.2 µm

Suspension polymerization is used to produce polymer suspensions with particle diameters larger than 0.2 µm. Suspension polymerization refers to the processes in which polymer dispersion particles are formed directly from monomer droplets of the initial emulsion [33,34,35,36,37,38,39,40,41,42,43,44,45,46,47]. In this case, the difference in the size distribution of polymer particles and initial emulsion droplets is attributed only to the coagulation of PMPs and to the Ostwald ripening process at the initial stage of polymerization [23].

In suspension polymerization, water-soluble polymeric surfactants (gelatin, starch, carboxymethyl cellulose, polyvinyl alcohol, acrylic monomer copolymers, etc.) are traditionally used to stabilize monomer droplets. These surfactants are much less surface active than the ones used for emulsion polymerization, and their concentration is some 10 times lower [34]. That leads to low stability of PMPs and their coagulation at the initial stage of polymerization, and it results in a suspension with larger, and sometimes nonspherical, particles with a wide distribution of the particle sizes. An increase in surfactant concentration leads to an increase in PMP stability; however, in the aqueous phase there appear some aggregates of polymer stabilizer molecules, which swell with monomer, capture radicals from the aqueous phase, and also transform into PMPs. The PMPs formed this way, according to [33,34], are ~0.2 µm in size and undergo emulsion polymerization with the formation of a high-molecular-weight polymer. The polymer dispersion particle-size distribution becomes extremely broad. Therefore, the synthesis of stable polymer suspensions with narrow particle-size distribution using the suspension polymerization method becomes problematic. Various possible mechanisms of PMP formation are complicated by particle coagulation that makes it difficult to regulate the particle diameter of polymer dispersions and the width of their size distribution. Only the use of special techniques, such as gradual seeding of the monomer (seed polymerization) makes it possible to obtain polymer suspensions with narrow PSD of large diameter [33].

In addition to the physical and chemical processes mentioned above, which determine the particle size of polymer dispersions and their size distribution, there is a dropwise of monomer droplets after the beginning of polymerization [45]. The reason for this unusual phenomenon is related to the fact that the fragmentation of monomer droplets occurs simultaneously with an increase in viscosity of the dispersed phase due to polymer formation and reduction in interfacial tension.

The considerable theoretical interest in the topochemistry and mechanisms of heterophase polymerization so far has been related both to the lack of a complete general picture of the process and to the presence of specific features inherent in these processes. There is no unified viewpoint on the mechanism of formation of adsorption layers on the surface of polymer monomer particles, whereas their thickness and strength do undoubtedly influence the stability of polymer suspension and the kinetics and mechanism of heterophase polymerization.

## 4. Synthesis of Polymeric Microspheres Stabilized with Water Insoluble Surfactants

It is possible to imagine a situation in which PMP formation occurs only from monomer microdroplets. In this case, in order to obtain polymer suspensions with a narrow size distribution, one has to eliminate the surfactants, in the form of micelles or polymer aggregates of molecules, as well as monomer microemulsification, in the aqueous phase.

To solve some of the abovementioned problems, the use of surfactants that are insoluble in aqueous medium has been proposed [46]. The use of such surfactants for the synthesis of polymer microspheres and for their stabilization will be possible if they provide not only the formation of polymer–monomer particles by the same mechanism, from monomer microdroplets, but also the formation of strong interfacial layers on their surface. Their use will also make the polymerization process environmentally friendly, as the wastewater will not contain any water-soluble ingredients. The main condition for their application is the formation of an oil-in-water emulsion. For this purpose, the surfactant must be able to form a structural–mechanical barrier on the surface of the particles to avoid coalescence of monomer droplets and to maintain the stability of the oil-in-water emulsion [47]. Certain semiproducts of basic organic synthesis were tried out as such surfactants, such as di-p-tolyl-o-carbaloxyphenylcarbinol (DTC) and monoester of aromatic dicarboxylic acids (MAF) [46]. Polystyrene and polymethyl methacrylate PMPs with a narrow size distribution, with diameters of 0.5–0.9 µm, were synthesized in the presence of DTC and MAF, but they were stable only when the monomer concentration in the polymer suspension did not exceed ~20–25%.

Among water-insoluble surfactants of particular interest are functional organosilicon co-surfactants [46,47,48,49]. The high flexibility of the polydimethylsiloxane chain should contribute to the formation of a structural–mechanical barrier in the surface layers of particles and to the formation of stable polymer suspensions.

The presence of terminal functional groups in the organosilicon chain expands the fields of application of polymer suspensions obtained in their presence. The co-surfactant functional groups are oriented at the interface during polymerization, thus providing an opportunity for further modification of polymeric particles.

In this review, the results of research into the use of a wide range of functional polydimethylsiloxanes as surfactants for the synthesis of stable polymer suspensions with narrow size distributions are presented.

## 5. Functionalized Organosilicon Surfactants, Their Colloidal–Chemical and Rheological Properties

This research area started with investigation of the polymerization of monomers in the presence of an organosilicon surfactant, α-(carboxyethyl)-ω-(trimethylsiloxy) polydimethylsiloxane (PDMS), with a narrow length distribution of [-OSi(CH_3_)_2_-] chains and terminal carboxyl groups HOOCCH_2_CH_2_Si(CH_3_)_2_[OSi(CH_3_)_2_-]n-OSi(CH_3_)_3_, n = 7 to 9 [46,49]. PDMS was characterized by a diphilic structure similar to that of anionic surfactants widely used in industry for the synthesis of polymers and polymer suspensions. The length of the hydrophobic part of the surfactant molecule was 7–9 dimethylsiloxane units, which corresponded to the optimum length of the hydrophobic radical of the anionic surfactant.

First, it was of interest to find out whether there are any fundamental differences in the colloidal–chemical properties of water-insoluble and water-soluble surfactants, in the rheological properties of surfactant interfacial layers, and in the kinetic and topochemical regularities of monomer polymerization in their presence.

As PDMS was supposed to be used as the surfactant for heterophase polymerization, it was appropriate to compare the colloidal–chemical properties of PDMS with those of polyvinyl alcohol (PVA), widely used in suspension polymerization, and sodium dodecyl sulfonate (SDS), used in emulsion monomer polymerization surfactants.

In [46,49] it was found that PDMS is a surfactant and reduces the interfacial tension at the boundary styrene solution PDMS/water to 25 mN/m.

The interfacial tension (σ_1,2_) isotherm is shown in Figure 1 [46], and the colloidal–chemical properties are given in Table 1. It can be seen that the values of maximum adsorption, Γ_max_, and surface activity, G, for PDMS and PVA are close.

The results suggest the possibility of obtaining a highly dispersed polymer suspension with narrow size distribution using heterophase polymerization of monomers if a strong interfacial layer is formed on the particle surface.

There are few data in the literature on the rheological and strength properties of interfacial surfactant adsorption layers at the boundary between the solution of the polymer in the monomer and the aqueous solution of the surfactant, which would emulate the PMP/water interface.

In order to obtain information about the rheological parameters of PDMS, the formation times of the interfacial adsorption layers were determined beforehand (Figure 2). It was shown that the time required for the formation of the adsorption layers was 40 min, after which the rheological properties were virtually unchanged. The rheological characteristics of the interfacial adsorption layers were measured using a surface elastoviscosimeter [24].

Shear stress development curves for the interfacial adsorption layer over time at different concentrations of PDMS formed at the water/xylene interface have a pronounced maximum, which can be regarded as the ultimate strength of the layer structure (Figure 3). It corresponds to the ultimate breaking stress *P_rs_*. The basic rheological parameters of PDMS are shown in Table 2 [24].

The appearance of a maximum on the curves is due to the formation of a structure with solid-like properties. The shear stress then decreases to a steady-state stress (plateau in the curves). The stress *P_ss_*, maintaining the stationary flow, characterizes the viscous properties of the layer [24].

Based on the data shown in Figure 3, the dependence of *P_rs_* and *P_ss_* on PDMS concentration were plotted (Figure 4). It can be seen that *P_rs_* increases with increasing polymer concentration and reaches its highest value of 23.9 × 10^−3^ mN/m at 1% PDMS. With further increases in PDMS concentration, this value is practically unchanged.

The stress maintaining the stationary flow, *P_ss_*, also increases with increasing concentration of PDMS, but after reaching a value of 6.6 × 10^−3^ mN/m at 1 wt % PDMS, it does not change any more.

From the curves shown in Figure 3, the rheological parameters of the interphase adsorption layer of PDMS formed at the water/xylene PDMS solution interface, were calculated using Equations (1) and (2).
(1)$$E=P_{s}/\varepsilon$$
(2)$$\eta =P_{ss}/\dot{\varepsilon }$$
where *ε* = strain in fractions of relative shear, $\dot{\varepsilon }$ = strain rate, *E* = modulus of elasticity, *η* = viscosity.

A complete rheological curve for the interfacial adsorption layer of PDMS at the water/m-xylene 2% PDMS solution interface is shown in Figure 5. Two yield points can be distinguished on this curve. Up to the first yield point, P_k1_, an elastic–plastic region is found in which the strains of the interfacial layers of the surfactant are reversible. The mechanical properties in this region are characterized by the elastic modulus of deformation (Hooke’s law). The reversibility of the deformation is ensured by a change in entropy associated with a change in the mutual orientation of the surfactant molecules in the interfacial layer. Under these conditions, the surfactant interfacial layer is characterized by thermodynamic and mechanical reversibility.

At stresses greater than P_k1_, at low flow velocities little damage occurs to the system, as fractures inextricably linked to the flow have time to recover in a thixotropic manner. In this case, the flow of the system occurs almost without structure failure, i.e., creep phenomena are observed. The magnitude of this slow creep with constant plastic viscosity is 12.5 × 10^−3^ mN∙s/m [24].

At the second yield strength, P_k2_, which is obtained by extrapolating the rectilinear section on the strain rate vs. shear stress curve (Figure 5) to the intersection with the abscissa axis, the surfactant structure in the interfacial layer cannot withstand stationary flow and collapses. Above the second yield strength, the interfacial adsorption layer flows with the minimum plastic viscosity, the Bingham (1.3 × 10^−3^ mN∙s/m). The presence of two yield strengths indicates the hardness of the formed structures.

Several types of conformations for linear chains of PDMS have been suggested in the literature [46,49,50,51,52,53,54,55,56,57,58,59,60]. Such structures have been established from the NMR ^29^Si and NMR ^13^C spectra of crystalline PDMS [61,62]. All three possible structures are shown in Figure 6.

The relationship between viscosity and shear stress is shown in Figure 5b. It is believed that the difference between the Shvedov and Bingham viscosity values within the same order of magnitude is due to an orientation effect in the flow process. In a study of shear stress development for all interfacial adsorption layers of PDMS at the water/xylene interface, at a constant strain rate, the shear stress first increased, then decreased, then increased again, etc. Thus, the interfacial adsorption layer of PDMS tends to recover, i.e., exhibit thixotropic properties in the flow. Figure 7a,b show the thixotropic properties of the interfacial adsorption layers formed by 1% and 2% PDMS, respectively, at the water/xylene interface (9). At the interface, a weak structure similar to conventional thixotropic-coagulation structural meshes emerges, in which the particles are bound by van der Waals forces acting between hydrocarbon hydrophobic groups of PDMS molecules in contact with water. In [62,63] it was assumed that a monolayer of oriented PDMS helices (so-called Damascene helices [59]) was formed at the interface.

Thus, presumably, a two-dimensional self-organizing structure of polydimethylsiloxane characterized by a liquid crystalline state, and possessing thixotropic properties, is formed at the water/surfactant in m-xylene interface. The thixotropic properties of the polydimethylsiloxane interfacial adsorption layer are essential, for they provide the “healing” of defects in the fractured layer, as well as increasing its resistance to fracture, i.e., a structural–mechanical barrier is formed [24].

The results of the study of the rheological properties of the interfacial adsorption layers of surfactants used in suspension and emulsion polymerization, such as SDS, PVA, and PDMS, are shown in Table 3. Significant differences in the rheological characteristics of interfacial adsorption layers formed by water-soluble and water-insoluble surfactants are evident.

A comparison of the rheological characteristics of the interfacial adsorption layers of surfactants was performed at their concentrations as commonly used in the polymerization process: 4% for SDS (shear stress limit 1.5 × $10^{-3}\;$mN/m), 1% for PDMS (shear stress limit 23.9 × $10^{-3}$ mN/m), and 0.5% for PVA (shear stress limit 0.95 × $10^{-3}$ mN/m). Thus, even when low concentrations of PDMS are used, it forms an interfacial adsorption layer of a higher strength compared with SDS and PVA used at higher concentrations. The formation of polymers on the surface of the monomer droplets will promote the hardening of the adsorption layer, therefore increasing the stability of the suspension particles from the early stages of polymerization. The results show that the rheological properties of the interfacial adsorption layers formed by water-insoluble PDMS and polymers are promising. These results are specific to this type of surfactant.

The conformational behavior of polydimethylsiloxane molecules in insoluble monolayers on liquid nonaqueous surfaces was investigated in [64]. It was shown that flexible linear polymer chains form a 2D random coil conformation.

The destruction of the monolayer by its compression results in the creation of a folded chain structure, with the macromolecules transitioning into a spiral conformation.

A model of the 2D structural organization of polydimethylsiloxane macromolecules is showed in Figure 8.

## 6. Polymerization of Styrene in the Presence of PDMS

The conversion curves for styrene polymerization in the presence of PDMS and, for comparison, in the presence of water-soluble surfactants normally used for emulsion and suspension polymerization (SDS, PVA, gelatin, and styrene polymerization in bulk), other conditions being equal, are shown in Figure 9. One can easily see that the shapes of the kinetic curves obtained in the presence of surfactants of different natures are almost identical.

The shapes of the kinetic curves are similar, but the dependencies differ from each other by the values for the polymerization rate and the values for the induction period corresponding to the time of formation of PMPs. The styrene polymerization rate in the presence of PDMS (curve 2) is 2.5 times higher than the mass polymerization rate (curve 5), close to that observed for polymerization in the presence of polyvinyl alcohol (curve 3). It is also two times lower than the emulsion styrene polymerization rate in the presence of SDS (curve 1) [61].

From the data shown in Figure 10, one can easily see that the only polystyrene suspensions truly characterized by a narrow particle distribution are those obtained in the presence of PDMS. Moreover, the diameter of the particles obtained in the presence of PDMS is approximately 10 times larger than that of the particles obtained in the presence of (water-soluble) SDS (Figure 10).

The average particle diameters obtained in the presence of PDMS remained practically unchanged up to complete monomer conversion, and the particle-size distribution remained narrow. The average diameter of polymer microspheres is 0.56 µm. Microphotographs of the particles and histograms of their diameter distribution are shown in Figure 11.

The effect of initiator concentration on polymerization rate, polymer molecular weights, particle diameter, and particle-size distribution is shown in Figure 12 and Table 4.

The rate of polymerization is proportional, and the molecular weight of the polymers is inversely proportional, to the initiator concentration to the power of 0.5, which is consistent with the theory of radical polymerization. The size of polymer particles in the suspensions is practically independent of initiator concentration and is characterized by a narrow size distribution. The stability of polymer suspensions is maintained if the initiator concentration is very low, that of 0.1 wt % styrene. The results are shown in Table 4.

The concentration of potassium persulfate (K_2_S_2_O_8_, PPS) was varied, with a wide range of values—from 0.1 to 4 wt % styrene—and the polymerization was carried out at a constant concentration of initiator PPS—1 wt % monomer and monomer surfactant/water ratio of 1:9. The results are shown in Table 5.

It should be noted that the stability of the polymer suspensions is already achieved at low concentrations of PDMS, as convincingly evidenced by the absence of coagulum in the suspension.

It was proposed that the stability of the polystyrene suspension is ensured mainly by the structural–mechanical stabilization factor due to the formation of a strong interfacial layer of PDMS and the polymer. The contribution of the electrostatic stabilization factor seems to us to be considerably less.

X-ray photoelectron spectroscopy showed that the surface of polystyrene particles contains 37 times more PDMS molecules than the volume. Therefore, the polymer–monomer particles (PMPs) must have a core-shell structure, in which the shell consists of PDMS, forced out by the polystyrene to the interface. The formation of the shell, consisting of PDMS molecules, provides the particles with certain properties, which are characteristic of organosilicon polymers, such as poor swelling in monomers. For example, average particle diameters of polystyrene suspensions obtained in the presence of 2% and 4 wt % PDMS (0.47 µm and 0.51 µm, respectively) increased by around 60% after swelling in styrene for 72 h.

Thus, a feature of polymer microspheres formed in the presence of PDMS is that the interfacial adsorption layer of the particles contains an organosilicon surfactant which, combined with the polymer, provides stability of the reaction system in the polymerization process, and hence no coalescence of particles and a narrow particle-size distribution.

The mechanism of interfacial layer formation with water-soluble micelle-forming surfactants is quite different. In order to obtain polystyrene suspensions with similar stability, it is necessary to use significantly higher concentrations of a water-soluble surfactant, which means that to obtain particles with a core-shell structure one has to carry out multistage seed polymerization. In the presence of water-soluble surfactants during the interface layer formation, one may observe a strict separation into two layers: hydrophobic, which is essentially hydrocarbon, and hydrophilic, formed by the polar groups. At the same time, organosilicon surfactants form a thick interfacial layer, in which there is a smooth transition from the hydrophobic part to the hydrophilic one, containing functional terminal groups and -Si-O-Si- groups. This is what determines the effectiveness of the stabilizing effect of the thin liquid crystalline film of organosilicon surfactant arising on the surface of PMPs and maintaining the stability of the particles against coagulation.

The possibility of creating a hydrophilicity gradient on the surface of the particles is determined by the chemical structure of the organosilicon surfactant. The presence of methyl groups in the surfactant molecule allows it to acquire a conformation that ensures their maximum concentration near the interfacial polymer surface, while the hydrophilic groups are oriented towards the aqueous phase.

The high stability of monomer emulsions has been attributed to the formation of a strong interfacial layer of organosilicon surfactant and a polymer incompatible with it. In order to confirm the incompatibility of surfactant and polymer, their mutual diffusion was studied by optical interferometry.

Typical interferograms of interdiffusion zones are shown in Figure 13. These results were obtained when polystyrene (PS) and polydimethylsiloxane phases of the dimeric structure PDMS(COOH) and ridge-shaped PDMS (PDMS G) came into contact. In the interferograms, the interface (F), areas of pure PDMS and PS, and the zone of dissolution of PDMS in PS can be clearly identified. Note that the diffusion of PS into PDMS is negligible. For comparison, we present interferograms for polyphenylmethylsiloxane Figure 13a,b, which show the diffusion front moving into the polystyrene region. For all organosilicon samples, during the experiment on mutual solubility of components (from 1 to 104 min) the interface was preserved. The diffusion zone thickness increased in time, which was accompanied by a concentration gradient decrease, while the quantitative characteristics of its composition did not change. This leads one to the conclusion that a diagram of amorphous stratification with an upper critical mixing temperature characterizes the system.

Note that the diffusion of PS into PDMS is negligible, and interferograms of interdiffusion of PS and PDMS without terminal-functional groups show a one-way movement of the PDMS front into the PS phase. With increasing temperature, the size of the PDMS diffusion zone in PS increases. The results show that the components of the mixtures studied are soluble to a limited extent in each other in the temperature range from 20 to 200 °C. It should be noted that the presence of carboxylic terminal groups, in the composition of PDMS without terminal-functional groups, positively affects the compatibility of the components of the polystyrene–PDMS system.

The distinctive feature of interfacial layer formation in the presence of organosilicon surfactants is the duration of its formation: this layer begins to form from the beginning of polymerization initiation, as a result of the polymer forcing out the surfactant to the surface layer, and it continues until the full conversion of the monomer is achieved. All the surfactant contained in the volume is concentrated in the interfacial layer, and this sets the stage for the formation of particles with a core-shell structure, which is confirmed by electron-microscopy data.

The results obtained in the presence of a co-surfactant were confirmed by data on the polymerization of styrene and MMA in the presence of water-insoluble oxyethylated polypropylene glycols, polyether, polylactides of different molecular weight, etc. All this testifies to the prospects of using surfactants of this type for the synthesis of PS with narrow PDD.

## 7. Conclusions

This review article expounds the experiment results and the novel concept of the radical heterophase polymerization of vinyl monomers in the presence of water-insoluble (in particular organosilicon) surfactants of various structures. Due to the ability to establish “thick” adsorption layers on the particle/water interface, the polymerization in those systems allows one-step synthesis of polymer suspensions having large particles and a narrow particle-size distribution and with no residual surfactant in the wastewater.

## Figures and Tables

**Figure 1 polymers-15-02464-f001:**
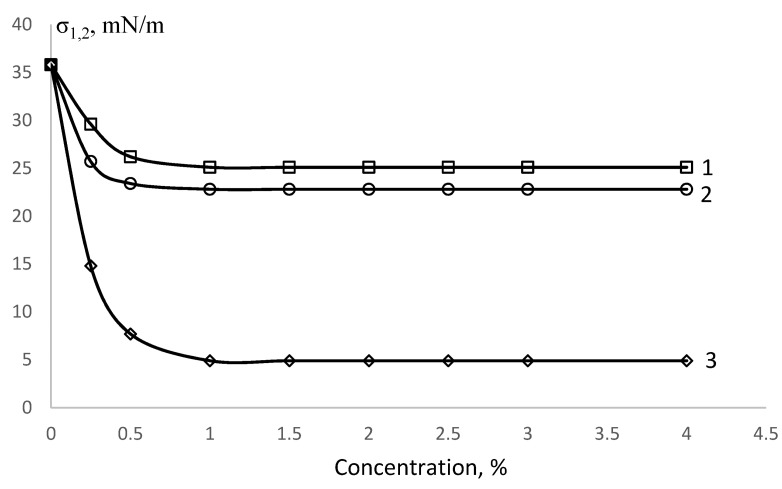
Isotherms of interfacial tension, phase boundaries: 1—styrene solution PDMS/water, 2—styrene/water PVA solution, 3—styrene/water solution SDS.

**Figure 2 polymers-15-02464-f002:**
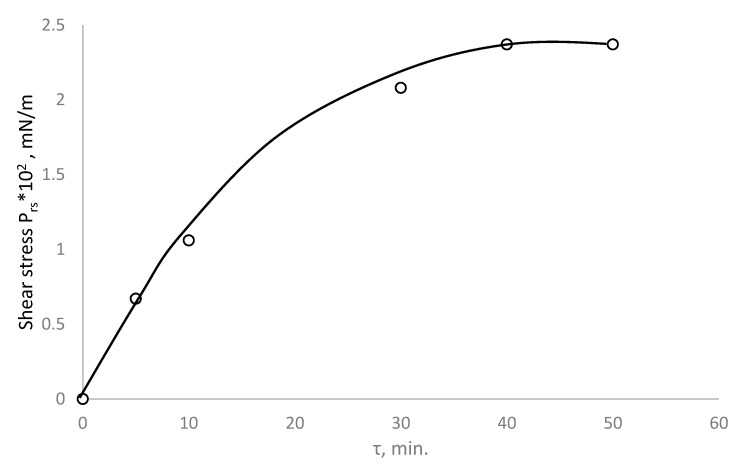
Dependence of ultimate shear stress *P_rs_* on the time of formation of the interfacial adsorption layer at the water/PDMS interface (1% PDMS in m-xylene) [24].

**Figure 3 polymers-15-02464-f003:**
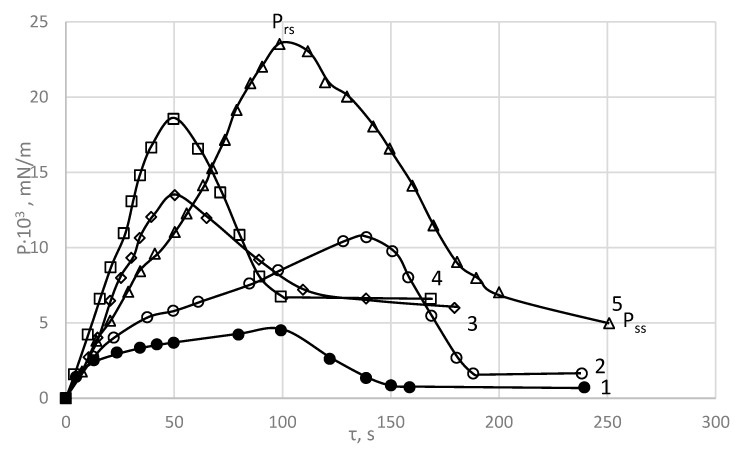
Time development of shear stress, P, at the water/1 wt % PDMS interface in m-xylene. Experimental temperature 20 °C, strain rate 0.92 s^−1^. Concentration of PDMS, vol %: 1—0.2, 2—0.6, 3—0.8, 4—1.0, 5—2.0 [24].

**Figure 4 polymers-15-02464-f004:**
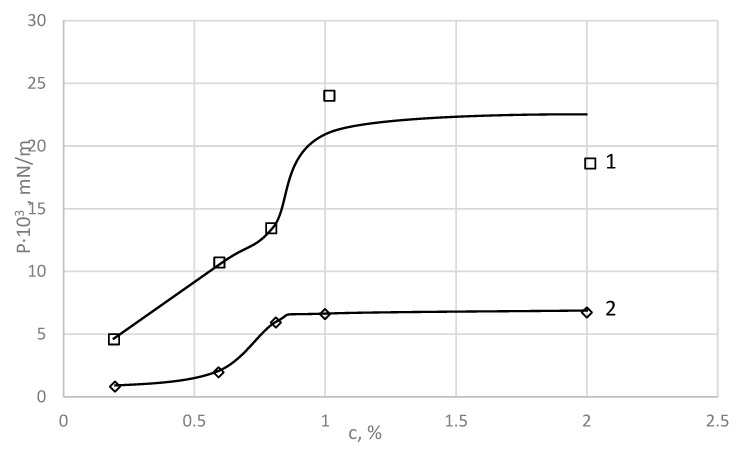
Dependence of critical shear stress, *P_rs_* (1), and stress maintaining the stationary flow, *P_ss_* (2), on the concentration of PDMS. Layer formation time = 40 min, experiment temperature = 20 °C [24].

**Figure 5 polymers-15-02464-f005:**
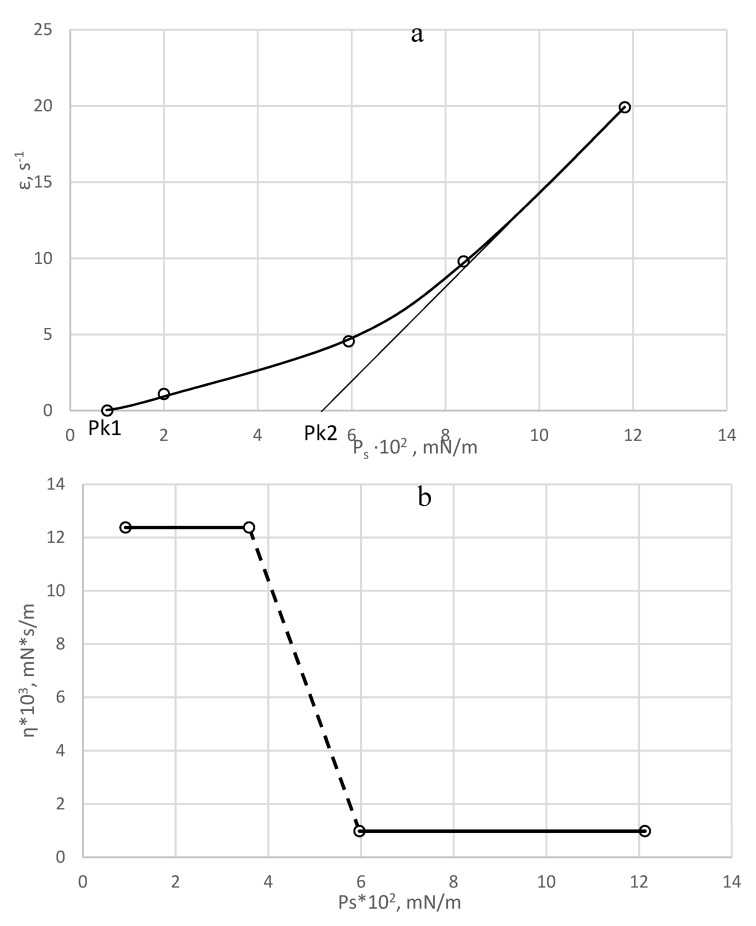
Dependence of strain rate (**a**) and viscosity (**b**) on shear stress at the water/2% PDMS in m−xylene interface [24].

**Figure 6 polymers-15-02464-f006:**
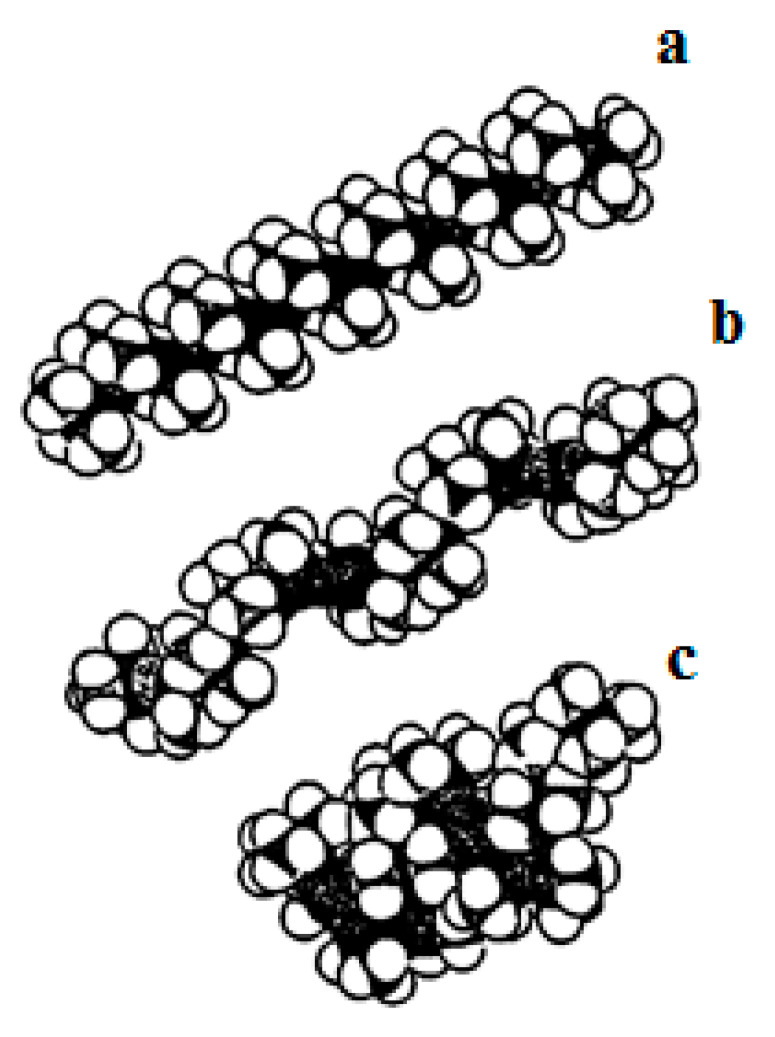
Possible conformational structures of PDMS chains: (**a**) cis-trans-conformation, (**b**) elongated helix, and (**c**) Damascene helix [24].

**Figure 7 polymers-15-02464-f007:**
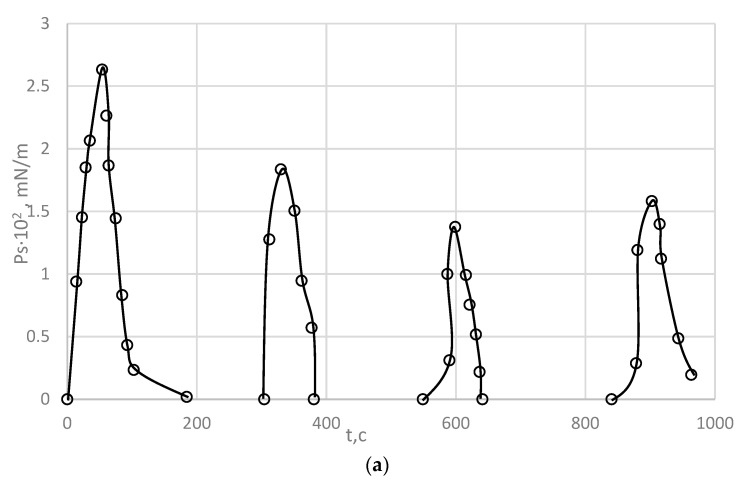
Thixotropic recovery of interfacial adsorption layer over time at the water/2% PDMS in m-xylene interface (**a**) and at the water/1% PDMS in m-xylene interface (**b**). The strain rates were 1.85 s^−1^ and 0.92 s^−1^, respectively.

**Figure 8 polymers-15-02464-f008:**
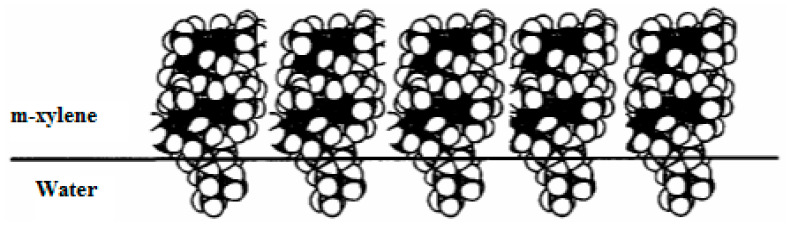
The location of PDMS molecules at the water/m-xylene boundary.

**Figure 9 polymers-15-02464-f009:**
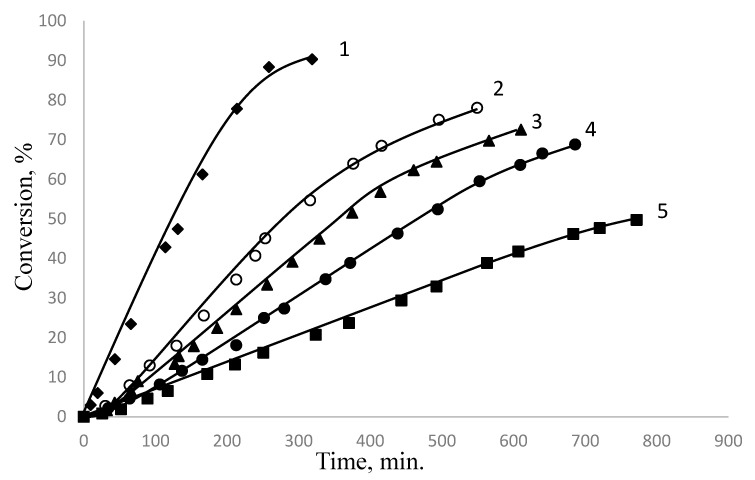
Conversion–time curves obtained by polymerization of styrene in the presence of 1 wt % K_2_S_2_O_8_ and various surfactants: 1—SDS, 2—PDMS, 3—PVA, 4—Gelatin, 5—polymerization in bulk.

**Figure 10 polymers-15-02464-f010:**
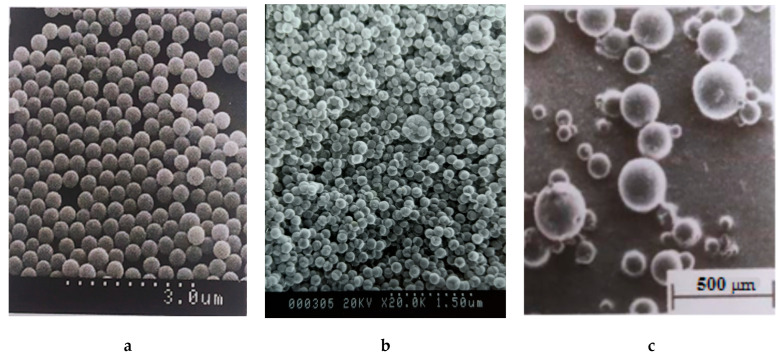
Microphotographs of polystyrene particles obtained in the presence of PDMS (**a**) [45], SDS (**b**) [46], PVA (**c**) [34].

**Figure 11 polymers-15-02464-f011:**
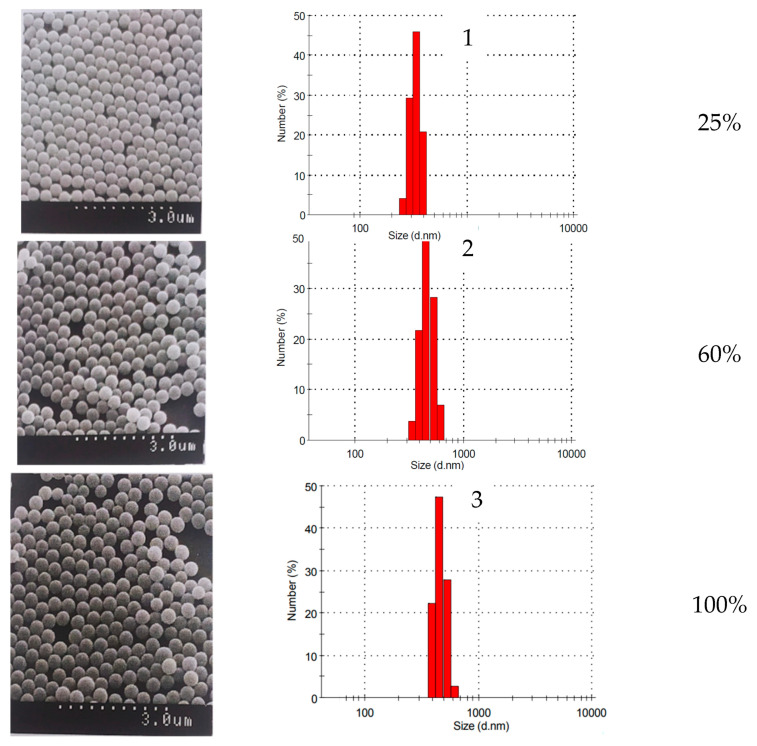
Electron micrographs of particles obtained at a volume ratio of monomer: water phases of 1:9. The concentration PDMS and K_2_S_2_O_8_ equal to 1.0 wt % per styrene. Monomer conversion: 1—25%, 2—60%, 3—100%.

**Figure 12 polymers-15-02464-f012:**
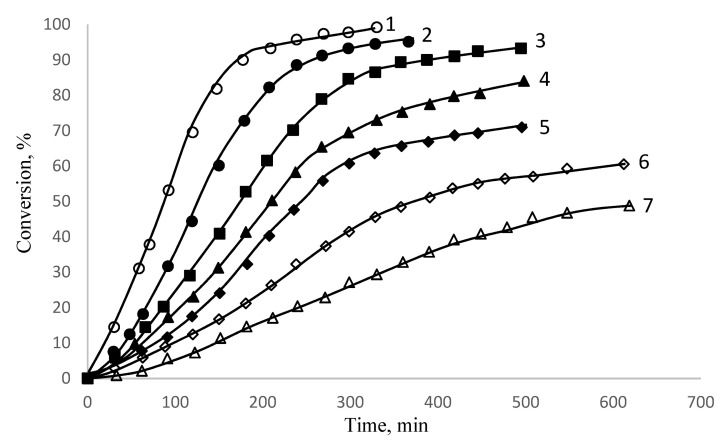
Conversion–time curves obtained at a volume ratio of styrene solution PDMS/water = 1:9, T = 70 °C, [PDMS] = 1 wt % per styrene and [K_2_S_2_O_8_]: (1) 4%; (2) 3%; (3) 2%; (4) 1%; (5) 0.5%; (6) 0.2%; (7) 0.1% (per styrene) [46,49].

**Figure 13 polymers-15-02464-f013:**
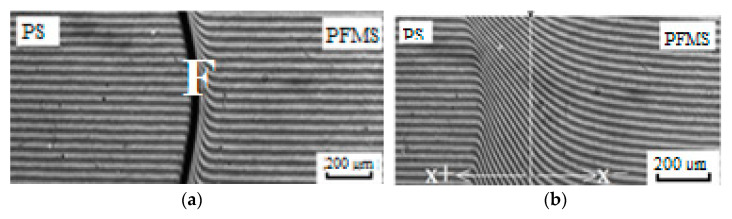
Interferograms of interdiffusion zones of the system PS4100-PDMS, where (**a**,**b**) PFMS at 160 °C, t = 1 min and t = 104 min, (**c**,**d**) PDMS(COOH) at 200 °C, t = 10 min and t = 30 min, (**e**,**f**) PDMS G at 180 °C, t = 0 min and t = 20 min.

**Table 1 polymers-15-02464-t001:** Colloidal–chemical properties of surfactants.

Surfactant	Solubility Ratio of Surfactants, Kv/Km	σ_1,2_, mN/m	Γ_max_ × 10^6^, mol/m^2^	G × 10^3^, mN∙m^2^/mol	S_0_, nm^2^
PDMS	3.0 × 10^−5^	28	10.7	33.4	1.55
PVA	-	25	14.4	29.1	1.29
SDS	-	4	4.5	14.5	0.40

**Table 2 polymers-15-02464-t002:** Rheological parameter of PDMS layer [47].

Rheological Parameters	PDMS Concentration, % (vol.)				
	0.2	0.6	0.8	1.0	2.0
*P_rs_* × $10^{-3}$, mN/m	4.5	10.6	13.3	23.9	18.7
*P_ss_* × $10^{-3}$, mN/m	0.8	1.8	6.1	6.6	6.6
Viscosity $\eta_{s}\times 10^{-3}$, mN · C/M	0.8	1.9	6.6	7.1	7.1
Modulus of elasticity $E_{s}\times 10^{-3}$, mN/m	2.5	4.0	5.2	9.8	9.8

**Table 3 polymers-15-02464-t003:** Rheological characteristics of interfacial adsorption layers of SDS, PVA, and PDMS at the interfaces of styrene/water.

Name	[Surfactant], %	$E_{s}\times \;10^{-3},\;\mathrm{mN}/m$	$P_{\mathrm{rs}}\times \;10^{-3},\;\mathrm{mN}/m$
40 kDa PVA, 2% content of acetate groups	0.01	1.2	0.6
	0.2	2.0	0.8
	0.5	3.0	0.95
SDS	2	-	1.12
	4	1.6	1.5
	8	-	10
PDMS	1	9.8	23.9

**Table 4 polymers-15-02464-t004:** Effect of potassium persulphate concentration on polystyrene suspension characteristics.

Initiator Concentration		Average Particle Diameter, $D_{n},\;µm$	$\mathrm{Polydispersity},\;D_{w}/D_{n}$	$\mathrm{Polymerization}\;\mathrm{Rate},\;w_{n}\times 10^{3},\;\mathrm{mol}/L$	Molecular Weight, $M_{n}\times 10^{-5}$
wt %	10^2^ mol/L				
0.1	0.33	0.40	1.02	1.3	7.0
0.5	1.68	0.41	1.01	3.8	3.7
1.0	3.36	0.43	1.01	4.3	2.4
2.0	6.70	0.43	1.01	5.5	2.1
4.0	13.40	0.44	1.01	11.2	1.8

**Table 5 polymers-15-02464-t005:** Effect of PDMS concentration on polystyrene suspension characteristics [46,49].

PDMS $\mathrm{Concentration}\times 10^{2}\;\mathrm{mol}/L\;\mathrm{Monomer}$	$\mathrm{Average}\;\mathrm{Particle}\;\mathrm{Diameter},\;D_{n},\;µm$	$\mathrm{Polydispersity},\;D_{w}/D_{n}$	Polymerization Rate, $w_{n}\times 10^{3},\;\mathrm{mol}/L$	Molecular Weight, $M_{n}\times 10^{-5}$
0.11	0.40	1.08	5.5	4.50
0.27	0.41	1.04	5.0	4.40
0.54	0.42	1.02	4.5	4.00
1.08	0.43	1.01	4.3	2.40
2.16	0.46	1.01	3.2	2.35
3.24	0.48	1.01	2.7	2.00
4.32	0.51	1.01	2.2	1.80

## Data Availability

Data is contained within the article.

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
