# Peer review of "New Approaches to the Synthesis and Stabilization of Polymer Microspheres with a Narrow Size Distribution"

_polymers, 2023, doi:10.3390/polym15112464_

Round 1

Reviewer 1 Report

This is a well written submission. The English is good with perhaps only minor grammatical amendments required. However, as a review article, I struggled at times to decipher whether the work being described is new (i.e., lack of appropriate citation), or taken from the existing literature. Further comments and examples are given in the following.

Throughout, check and use, eg 10.7, rather than 10,7 for decimal points.

Many references are Russian, and are either not readily available or not in translation.

A glossary of terms and symbols would be helpful.

Line 49: ... according to Harkins, Yurzhenko and Smith-Ewart ...

Line 52: ... the aqueous phase it enters into ...

Line 66: quasi-spontaneous

Line 141: I do not know what "n.tetradien" is

Line 149: delete co-surfactant, as it is unnecessary

Line 150: polydimethylsiloxane chains.  The high flexibility of polydimethylsiloxane chains ...

Figure 1: y-axis units - be consistent. Surface/interfacial tension is quoted as mN/m (as given in line 184)

Line 187: 10.7 x 10-6 etc, rather than 10.7·10-6

Line 188: Anionic surfactant ...

Line 202: Prs (and other instances elsewhere) should be Prs

Figure 2: As an example of the confusion between (possible) new results or literature results, there should be a citation here. Is the data here, for example, taken from reference 79?

Figure 3 and Table 2, no citations given. New data?

Likewise Figure 4 and the following section, where no references are indicated. New work?

Line 247: the Swede?? Not a term I am familiar with in this context

Line 254: Bingham fluid or model? Plastic viscosity? 1.3 x 10-3 not mN/m??

Lines 261-262: I confess to be unaware of this. Again, no citation given here

Figure 6: where is this taken from? Not specifically referenced

Table 4: Russian units given. Needs checking and referencing

Line 291: define MAC

Figure 7: from ref. 58? If so, it's in Russian

Figure 9: taken from where?

Lines 343-344: research from where?

Line 354: suspensions

Line 358: X-ray photoelectron spectroscopy is more usual name

Lines 365-367: these increases in diameter correspond to a doubling in particle volume, and so IS significant

Line 374: concentrations

Line 393: ... zones are shown ...

Figure 11: sources should be shown in the figure legend

Line 413: oC not Co

Overall, therefore, this submission needs to be revised to improve its format as a review article. The textual referencing needs to be greatly improved. If any of the data included here are new, then appropriate experimental details must be included.

Author Response

Dear Reviewer, The authors would like to thank you for reading the manuscript and for your valuable comments. useful for its improvement. All edits have been made to the text of the article.

Reviewer 2 Report

Author investigationed on heterophase polymerization of vinyl monomers in the presence of organosilicon compounds of different structure. Author was presented manuscript very well and its need minor revision. I have some comments:

1) Author should have given the temperature effect for used polymer with relevent references

2) What would be the additive effect on stabililty of polymer?

Author Response

Dear Reviewer, The authors would like to thank you for reading the manuscript and for your valuable comments. useful for its improvement. All edits have been made to the text of the article.

Polymerization conditions are given in articles:

Synthesis of polymer microspheres of different diameters in the presence of carbofunctional organosilicon surfactants / I. A. Gritskova, A. A. Ezhova, N. A. Lobanova [et al.] // Colloid & Polymer Science. – 2021. – DOI 10.1007/s00396-020-04805-2. – EDN IGELIW. 

Organosilicon Comb-Shaped Surfactants for the Synthesis of Polymer Suspensions with a Narrow Particle Size Distribution / A. A. Ezhova, I. A. Gritskova, M. A. Lazov [et al.] // Polymer Science, Series B. – 2021. – Vol. 63, No. 3. – P. 209-217. – DOI 10.1134/S1560090421030052. – EDN XCMGXS. 

Effect of the composition and structure of carbofunctional oligodimethylsiloxanes on their colloidal-chemical properties / I. A. Gritskova, A. A. Ezhova, A. E. Chalikh [et al.] // . – 2019. – Vol. 68, No. 1. – P. 132-136. – DOI 10.1007/s11172-019-2428-0. – EDN YUOLWB. 

Behavior of Organosilicon Surfactants in Langmuir Films on the Surface of Water / A. A. Ezhova, I. A. Gritskova, S. N. Chvalun [et al.] // Polymer Science, Series A. – 2019. – Vol. 61, No. 2. – P. 149-156. – DOI 10.1134/S0965545X19020044. – EDN HNKRVP. 

Round 2

Reviewer 1 Report

The authors have not clearly spelt out the changes made in a specific cover letter. They said that they have made the suggested changes, but not given details, which has therefore required further detailed reading of the amended manuscript necessary. However, I see that my suggested changes have mostly been made, but the use of commas rather than decimal points in numeric data has not been fully amended. This needs to be re-checked and commented on.

Author Response

Dear Reviewers!

Thank you very much for the detailed highly professional analysis of our manuscript and valuable suggestion. We made corrections of the text at all positions indicated in your review and in he letter of the 3rd May. The corresponding corrections are marked yellow. The suggestions of the other reviewer will be taken into consideration in the following publications.

With best regards

Sincerely

I.A. Gritskova
